# A Cost–Benefit Approach to Assess the Physical and Economic Feasibility of Sand Bypassing Systems

Márcia Lima [1,2,*], Ana Margarida Ferreira [1] and Carlos Coelho [1]

1   RISCO & Civil Engineering Department, University of Aveiro, Campus Universitário de Santiago, 3810-193 Aveiro, Portugal; margarida.ferreira@ua.pt (A.M.F.); ccoelho@ua.pt (C.C.)
2   Porto University Center (CUP), Lusofona University, 4000-098 Porto, Portugal
*   Correspondence: marcia.lima@ua.pt

**Abstract:** The artificial sand bypassing systems are coastal interventions designed to transfer sediments from areas of accretion to areas where erosion is observed. The goal of these systems is to reduce the littoral drift gradients where sediment transport is interrupted (vicinity of river mouths or port structures) and maintain a balanced sediment distribution along the shoreline. However, these systems present high initial investment costs as well as ongoing expenses for operation and maintenance. To assess the feasibility of sand bypassing systems in mitigating coastal erosion from a long-term perspective (decades), a comprehensive understanding of their performance is necessary, considering both physical and economic aspects. Thus, a cost–benefit assessment numerical tool, COAST, is applied to evaluate and discuss the effectiveness of bypassing systems during their life cycle. First, a comprehensive analysis is conducted for a generic study area, and then, the feasibility of the systems is discussed for two real study areas of the Portuguese West coast (Barra-Vagueira and Figueira da Foz-Leirosa). The generic results demonstrate as the importance of systems location or transposed flow volumes. On the other hand, the analysis of the two real case studies revealed that, under similar conditions, the results of the cost–benefit analysis can be contradictory. For the Barra-Vagueira stretch, the work indicates that the bypass system is not economically viable, whereas for Figueira da Foz-Leirosa, it is cost-effective. The study shows the importance of the site-specific conditions to evaluate the best option for a medium to long-term planned coast, highlighting the relevance of the proposed approach to assess the physical and economic feasibility of sand bypassing systems.

**Keywords:** LTC; COAST; sediments balance; coastal management; Portuguese Northwest coast

## 1. Introduction

The jetties implemented in coastal inlets have the objective of stabilizing the shoreline position and ensuring adequate navigation conditions inherent to the maritime activities [1,2]. The main impact of these structures is on longshore sediment transport, as they interrupt the natural littoral drift, leading to morphological changes in coastal zones [1,3,4]. A common observation of the long-term impact of interrupting the natural sand transport is the accretion of the updrift beach and erosion effects downdrift. Artificial sand bypass solutions are measures adopted to minimize the negative effects of the structures downdrift, aiming to restore natural sediment transport and mitigate the sediment deficit caused by the structures [5–7]. According to the data collection presented by Boswood et al. [3], Klein [7] and Coelho et al. [8], fixed sand bypassing systems have been implemented in various locations worldwide, such as in the USA (South Lake Worth Inlet, Port Sanilac and Lake Worth Inlet and Oceanside Harbour), Australia (Nerang River Entrance and Tweed River Entrance), South Africa (Baía de Algoa) and Brazil (Barra do Furado).

The sand bypassing components include dredging, transportation and discharging systems and can be carried out using different methods. Loza [9] states that these systems

can be classified based on one or a combination of several parameters, including purpose, mobility/flexibility, operation mode, operation schedules and capacity. The most common solution for sand bypassing is a mobile system that utilizes dredgers to recover the sand combined with boats and/or submerged temporary pipelines and booster pumps to transport the sediments and deposit them on downdrift beaches or in the nearshore zone [6,10]. These systems can be relocated to reach various areas of the bypassing site. Another method to perform the sand bypassing involves implementing a bypassing plant in a set location [3,6,10]. The fixed systems are typically composed of a jetty supporting sand recovery equipment and permanent pipelines with intermediate booster pumps [6,9]. A key parameter for a successful sand bypassing design for a specific site is the sediment transport volume capacity, but the project should also include other aspects, such as coastal processes characterization, waves and currents, bathymetry and topography, river and stream outflows, evolution of the coastal systems, storm impacts, social factors and environmental constraints [7,9]. While it is not within the scope of this work, it is important to acknowledge that sand bypassing systems have associated environmental impacts that demand management and monitoring during both the construction and operational phases of the system [11,12].

An important aspect of bypassing systems is the economic charge necessary to ensure the adequate functioning of the system throughout the defined project lifecycle. Bypassing costs for mobile systems are generally determined through navigation dredging contracts. Estimating costs for a fixed system must include the construction of the bypass system, operational expenses (such as electricity, salaries and wages), repair and maintenance costs and the dismantling costs of the system at the end of the lifecycle [3,8]. Clausner [10] suggests that the costs and performance of a fixed system should be estimated to allow for a comparison of costs and benefits with the traditional method of sand bypassing (dredging and deposition). This facilitates the identification of the optimal method for inlet maintenance. Cost–benefit assessment helps to predict and compare the economic performance of a project, combining the monetary values by the evolution of the benefits and costs of a project throughout its lifecycle and updating to their present value using a discount rate [13,14]. As a result, information regarding the viability of a project based on economic parameters, such as net present value (NPV), benefit–cost ratio (BCR) and break-even year, is obtained. A project is considered viable when the NPV is higher than zero and the costs are compensated by the benefits when the BCR > 1 [14,15]. Although it is not within the scope of this work, it should be noticed that the definition of land values for the benefit evaluation encompass at the same time economic, social, cultural and environmental aspects and require a sensitivity analysis prior to its application to adequate characterize the provided services of the territory. The objective of this work is to discuss the physical and economic feasibility of fixed sand bypassing systems using a cost–benefit analysis designed to coastal projects. The study follows the methodology presented in Coelho et al. [15], which consist of three sequential phases: 1st—Assessment of the benefits of the interventions based on the evolution of the shoreline position obtained through a one-line numerical model; 2nd—Assessment of all the costs associated with the system throughout its lifecycle; 3rd—Cost–benefit analysis based on economic parameters NPV, BCR and break-even year. The study was initially conducted in a generic study area to provide a comprehensive understanding of the methodology and its potential applications. Subsequently, the feasibility of sand bypassing is discussed for two specific study areas located on the Portuguese West coast (Barra-Vagueira and Figueira da Foz-Leirosa).

## 2. Methodology

The methodology encompassed three phases to evaluate the physical and economic performance of diverse coastal interventions based on COAST tool [14,15]: (1) assessment of the intervention's benefits based on the physical performances of the systems over time and in terms of the areas maintained, gained or lost, obtained through results of a shoreline evolution model; (2) estimation of all the costs of the systems throughout their lifecycle,

considering construction, operation and maintenance and (3) cost–benefit analysis through economic parameters (net present value, NPV, benefit–cost ratio, BCR and break-even year), allowing us to discuss the feasibly of the intervention and supporting the decision making process. The NPV evaluation criterion was given by the sum of discounted benefits minus the sum of discounted costs that occurred in each period over the lifetime of the project (Equation (1), [13]). The BCR evaluation criterion was given by the sum of discounted benefits relative to the sum of discounted costs that occurred in each period t over the lifetime of the project (Equation (2), [13]). In turn, the break-even year corresponded to the year that the benefits were compensated by the costs. The investment was considered economically viable when the NPV > 0 or when the BCR > 1, i.e., when the present value benefits exceeded the present value costs. Note that the BCR = 1 when the NPV = 0.

$$\text{NPV} = \sum_{t=0}^{T} \frac{B_t}{(1+r)^t} - \sum_{t=0}^{T} \frac{C_t}{(1+r)^t} \tag{1}$$

$$\text{BCR} = \sum_{t=0}^{T} \frac{B_t}{(1+r)^t} / \sum_{t=0}^{T} \frac{C_t}{(1+r)^t} \tag{2}$$

This approach was in accordance with the works presented by Lima et al. [14] and Coelho et al. [8,15], which evaluated other coastal erosion mitigation interventions.

In the next two sections, this methodology is applied in two situations: a hypothetical case study (Section 3) and two real case studies (Section 4). First, a comprehensive analysis was conducted for a generic study area, discussing the feasibility of the systems based on several design parameters, such as the volume of sand transposed annually and the number and the location of the outlet(s). This generic study applied in a hypothetical area with simple characteristics allowed for the control de variables and understanding relevant aspects of the simulated scenarios. Second, the feasibility of the systems was discussed for two real study areas of the Portuguese West coast where sand bypassing systems were commonly referred to as a possible solution to mitigate the erosion effects that occurred downdrift of the harbour structures, which defined northern boundary of the study areas (Barra-Vagueira and Figueira da Foz-Leirosa).

For the generic study area, the reference scenario adopted represented the natural shoreline evolution, without any coastal intervention, in a controlled domain. Then, to show the use of the COAST tool, 4 scenarios were presented, starting at a baseline scenario definition that allowed for the discussion of the importance of location of the sand-bypassing outlets and sediment flow capacity. For the real case studies, the coastal stretch characterization was presented, followed by an estimate of the costs of the system (construction, maintenance and operation) and benefits (land use values and ecosystem services). The intervention scenarios were presented and their feasibility discussed, including a sensitive analysis to adopted assumptions. Notice that no detailed evaluation of the adopted monetary values is presented in this work, as economic values change in time and location, demanding adequate evaluations for future similar studies. Main importance was given to the type of costs and benefits considered in the methodology.

## 3. Generic Study Area

A generic reference scenario was generated, featuring a regular topo-bathymetry, depicted by a square grid of points, spaced 20 m. This grid consists of 401 points in the cross-shore direction and 501 points in the longshore direction, leading to a spatial domain area of $8000 \times 10{,}000 \ \text{m}^2$. The bathymetry was created following the Dean profile [16], using the parameters $m$ and $A$, which were 2/3 and 0.127, respectively. For the topography (above the reference water level, 0.0 m), a constant slope of 2% was defined.

The wave climate remained constant throughout all the numerical modelling simulations, which considered an offshore wave height ($H_0$) of 2 m, wave period of 9.34 s ($T$) and a wave direction of 10 degrees rotated from West, clockwise ($\alpha_0$). The active cross-

shore profile was limited by both the depth of closure ($DoC$ = 8 m) and the wave run-up ($R_u$ = 2 m), which resulted in a total active profile height of 10 m (constant along all the simulation). At the northern boundary of the domain, no sediment input was allowed (simulating an interruption on the littoral drift going in the simulated domain) and in the southern boundary, an extrapolation of the longshore sediment transport nearby was utilized. A time-step of one hour and a time scale of 20 years were adopted in all scenarios. Annual shoreline position outputs were recorded allowing for the yearly evaluation of the eroded areas.

To estimate territory value, the cluster of services of the urban areas and ecosystems that are important to human well-being, health, livelihoods and survival were considered. Similar to the works performed by Lima, Coelho and others [14,15], in this study, three distinct zones were defined, each with a constant landward value (Figure 1). From north to south, the study area encompassed beaches, an urban area and forests. The highest value was assigned to the urban area, covering a longshore extension of 1.5 km. The beach provides coastal protection and recreational uses, as the urban area supports several activities and uses (restaurants, hotels, stores, etc.), and finally, the forest provides climate regulation, timber, habitat for biodiversity, erosion control and many others [17,18]. It is important to understand that the defined land values include at the same time economic, social, cultural and environmental aspects and require previous sensitivity analysis on its application due to adequate characterization of the provided services within the territory. In spite of the hypothetical case study, the assumed values for this scenario are inspired and in accordance with common values in use for the Portuguese Northwest coast [8,15].

Nevertheless, in specific studies, users should provide support and validation for the values using appropriate socioeconomic databases, considering all the relevant aspects in the study site. For the reference scenario, a time discount rate (r) of 3% was adopted [14].

Taking into account the physical performance of the reference scenario, the observed shoreline evolution indicates significant erosion problems over the 20 years. This suggests that without interventions, the shoreline could recede about 230 m in the northern boundary of the study area, leading to erosion affecting the entire urban waterfront extension. Figure 1 illustrates the shoreline's position after 5, 10 and 20 years, along with the total lost area in each distinct zone.

The economic analysis relies on the unit values assigned to each coastal zone (€/m²/year), Table 1. The NPV was calculated at the end of each simulation year (while the BCR was not calculated, as no interventions or associated costs were involved in the reference scenario). Following five years of simulation, erosion and land losses amount to approximately 0.8 million euros. After 10 years, the costs surpass 3 million euros (values updated to year 0). At the end of the simulation, the land losses account for roughly 12 million euros.

**Table 1.** Economic land value considered in the case study (in accordance with [14]).

|        | Description | Location     | Extension (km) | Value (€/m²/Year) |
|--------|-------------|--------------|----------------|-------------------|
| Zone 3 | Beaches     | North limit  | 1.0            | 2.00              |
| Zone 2 | Urban area  | Intermediate | 1.5            | 10.00             |
| Zone 1 | Forests     | South limit  | 7.5            | 0.20              |

Even though hypothetical, the reference scenario demonstrates that in coastal areas susceptible to erosion (where the available sediments for the littoral drift are insufficient to match the potential sediment transport capacity and shoreline retreat rates are expected), significant economic losses will arise due to the direct reduction of areas providing land uses and ecosystems services. In fact, if no mitigation strategies are implemented, the loss of urban waterfronts, beaches and forests will diminish the benefits, and thus, a sand bypassing system is proposed as a solution to mitigate these erosion problems.

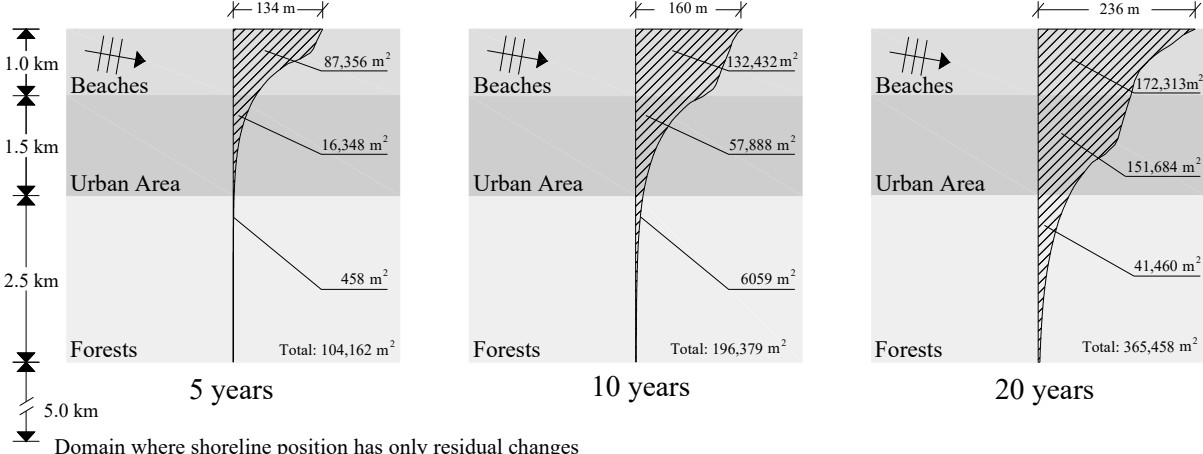

**Figure 1.** Shoreline position in the reference scenario, over the time (with a cross-shore scale 10 times larger than the long-shore scale).

### 3.1. Sand Bypass Baseline Scenario

The sand bypass baseline scenario, denoted as BS (Figure 2), featured a stationary structure positioned at the upper limit of the urbanized zone (located 1 km from the northern border of the spatial domain) once it is the highest territory value (to promote sediments accumulation in this richer area). This structure was considered to incur in an initial cost of 3 million euros, representing the sand bypass system costs. The sediments flow transposed by the system was presumed to satisfy approximately 90% of the potential wave climate sediment transport capacity at the beginning of the reference scenario simulations. The estimate was calculated by applying the CERC formula [19], resulting in 25 m$^3$/h or approximately 219,000 m$^3$/year. The cost of each cubic meter (m$^3$) of transposed sediments was estimated at 1 €/m$^3$, encompassing both operational expenses and maintenance costs for the bypass structure system. All the costs referred to in this study were considered in a generic perspective, although adequate for the Portuguese Northwest coast. They also allowed us to gain expertise on considering costs for the real cases presented in Section 4.

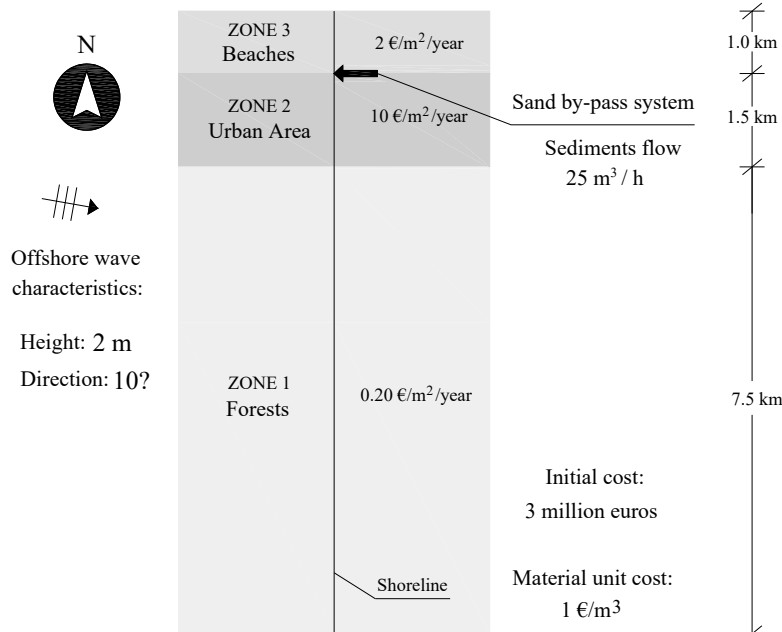

**Figure 2.** Schematization of the bypass baseline scenario.

In this scenario, shoreline evolution reveals smaller retreat rates near the northern border, compared to the reference scenario (Figure 3). However, erosion is still observed within the spatial domain, which can be attributed to the lower rate of transposed sediment flow in comparison to the potential sediments transport capacity. As a result of the chosen location for the transposition system, no shoreline retreat is observed along the urbanized zone throughout the 20-year simulation period. This coastal defence intervention is physically appealing as it not only reduces land losses (resulting in a positive impact of approximately 29 ha compared to the reference scenario), but also provides complete protection to the urbanized water front.

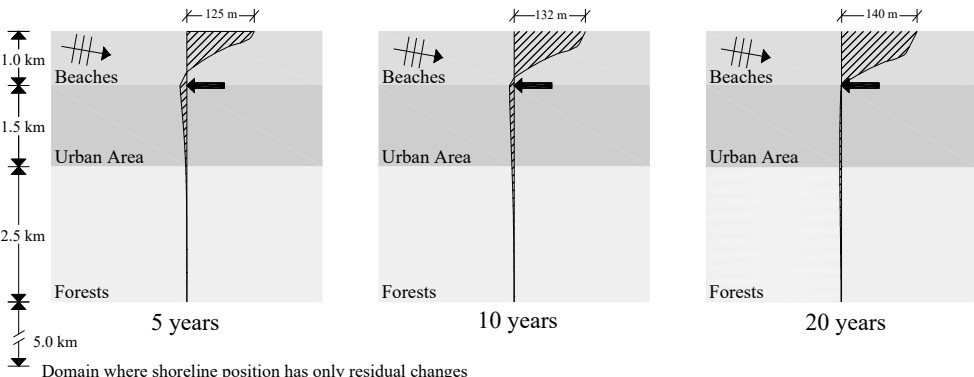

**Figure 3.** Shoreline position in the bypass baseline scenario along the time (with a cross-shore scale 10 times larger than the long-shore scale).

The costs associated with this scenario comprise the initial investment required to install the fixed sand bypass transposition system (estimated at 3 million euros) and the ongoing operation and maintenance costs, which were assumed to amount to €219,000, annually.

### 3.2. Interventions Scenarios

Two groups of scenarios were established to assess the main characteristics of the sand bypass, involving its location and bypassed sediments flow capacity (group *i* and *ii*, comprising one and three scenarios, respectively), as shown in Table 2. One scenario (*i*.1) was considered, in which the bypass is positioned 500 m north of the baseline scenario. Additionally, three scenarios (*ii*.1, *ii*.2 and *ii*.3) were defined to evaluate the capacity of the sediments flow rates. Among these scenarios, two have lower flow rates than the baseline, representing 10% and 50% of the wave climate sediments transport capacity, respectively, *ii*.1 and *ii*.2, while the third scenario has a 50% higher capacity than the wave climate sediments transport capacity (*ii*.3).

**Table 2.** Characteristics of coastal erosion mitigation scenarios.

|  |  | 1 | 2 | 3 |
|---|---|---|---|---|
| Location | *i* | 500 m from the northern border | - | - |
| Sediments flow | *ii* | 10% transport rate | 50% transport rate | 150% transport rate |

Table 3 shows the physical positive impacts of the bypassing system by evaluating the territory areas in each scenario. The impact is considered the difference of areas obtained in the analysed scenario and the reference scenario, depending on the shoreline position at the end of the simulation. It also shows the positive economic impacts, by summarizing the indexes obtained, which analysed the influence of the sand bypass system location and sediments flow capacity.

**Table 3.** Overview of the physical and economic outcomes of the sand bypass scenarios.

| | Scenario | Territory Area (m²) | | | BCR$_{20\,yr}$ (-) | NPV$_{20\,yr}$ (€) | Costs | | Break-Even (Years) |
| | | Accretion | Erosion | Impact * | | | Initial (€) | Total ** (€) | |
|---|---|---|---|---|---|---|---|---|---|
| *BS* | Figure 3 | 9946 | 84,454 | 290,950 | 1.89 | 5,548,634 | 3,000,000 | 6,258,167 | 13 |
| *i*.1 | 500 m from the northern border | 85 | 65,055 | 300,487 | 1.80 | 4,912,989 | 3,000,000 | 6,258,167 | 14 |
| *ii*.1 | 10% transport rate | 0 | 311,298 | 54,160 | 0.73 | −977,698 | | 3,625,568 | - |
| *ii*.2 | 50% transport rate | 63 | 197,531 | 167,989 | 1.61 | 2,930,383 | 3,000,000 | 4,824,574 | 15 |
| *ii*.3 | 150% transport rate | 119,620 | 6099 | 478,979 | 2.07 | 9,211,946 | | 8,630,113 | 12 |

* Impact value (m²) is obtained by the comparison of the accretion and erosion areas between the analysed scenario and the reference scenario (Figure 1). ** Values updated for initial simulation instant, according to the discount rate (r).

The sand bypass baseline scenario was designed to primarily safeguard the urbanized zone. Table 3 facilitates a comparison with a northern outlet location, revealing positive physical impacts from locating the bypass system to the north (yielding a benefit of approximately 1 hectare). However, from an economic standpoint, this is not the most optimal solution. The baseline scenario location yields higher BCR ratios and reaches equilibrium one year before scenario *i*.1.

As anticipated, a higher sediment flow transposed by the system leads to reduced land losses (Table 3). When compared with the reference scenario, scenarios *ii*.1, *ii*.2 and *ii*.3 avert losses of approximately 5, 16 and 48 ha, respectively. The economic outcomes of the scenarios align with this trend, as the scenarios with greater capacity also exhibit more favourable economic indexes. Within the 20-year time horizon, scenario *ii*.1 does not generate any monetary gains, whereas scenario *ii*.3 emerges as the most appealing option, both in terms of physical and economic benefits, with the benefits surpassing the total costs by more than twice after the 20 years (BCR = 2.07). In this scenario, accretion was attained, increasing the area by approximately 11 hectares compared to the initial moment. In conclusion, it was observed that the economic performance of the bypass systems improves with higher transposition capacity while it proves ineffective when considering lower sediment flows.

The original location of the sand bypass system in the baseline scenario was at the northern boundary of the urbanized zone. However, relocating the system 500 m northward leads to enhanced physical impacts, with reduced erosion. Increasing the capacity for bypassed sediments results in improved physical and economic indexes.

## 4. Case Studies: Barra-Vagueira and Figueira da Foz-Leirosa

Over the past few decades, the shoreline position downdrift of the Douro River mouth (Figure 4) on the West coast of Portugal has shown a retreat trend, leading the coastal management entities to adopt measures to mitigate the negative effects of coastal erosion, such as land loss, coastal floods and overtopping events [20]. This coastal area is part of the sedimentary cell of the Douro River—Nazaré, characterized by low topography with dunes, long and continuous beaches and exposure to wave actions from the Atlantic Ocean, with an average wave height of 2 m and wave directions predominantly from W and NW, promoting a littoral drift estimated at approximately $1 \times 10^6$ m³/year, processed from north to south [20,21].

According to Coelho et al. [8], the erosion phenomenon in the coastal areas located downdrift of the Douro River mouth is attributed to a sedimentary deficit caused by a reduction in sediments from the Douro River. This decrease is a result of actions carried out in the Douro River hydrographic basin over the years, including dam construction and sediments extraction. Additionally, anthropogenic actions carried out in the coast disrupt the natural processes of sediment transport in the coast, with consequences in the ability to replenish the littoral drift [20].

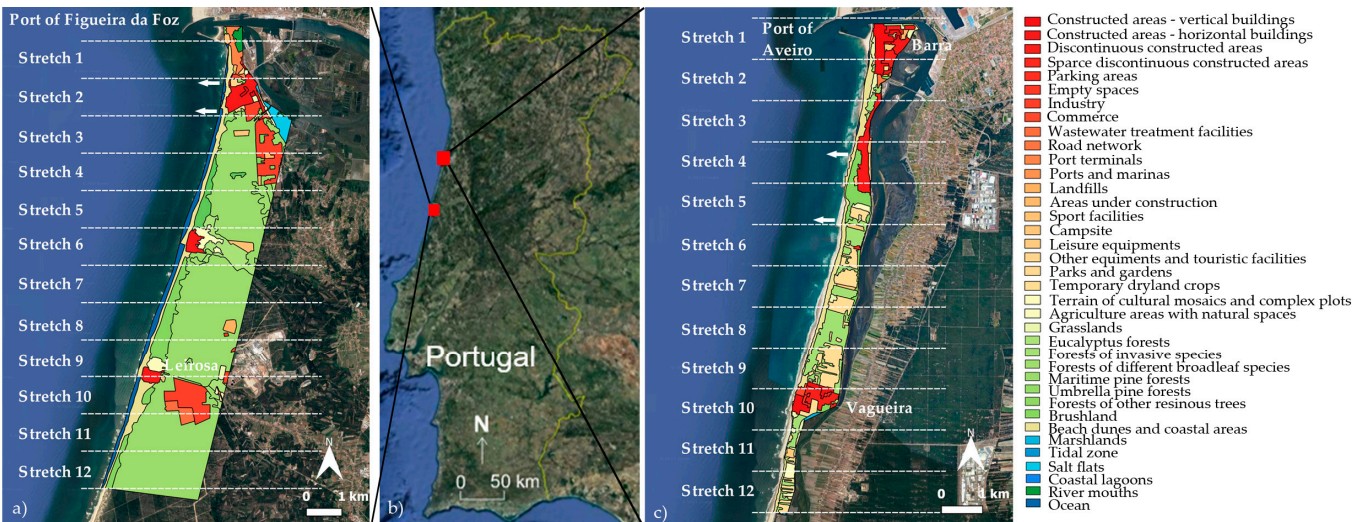

**Figure 4.** Areas of study with overlapping cartography of land use and the identification of the location of transposition system outlets (arrows): (**a**) Figueira da Foz-Leirosa; (**b**) Mailand Portugal; (**c**) Barra-Vagueira.

The coastal sectors Barra-Vagueira (Aveiro) and Figueira da Foz-Leirosa (Figueira da Foz) are among the areas along the Portuguese West coast that have been heavily affected by erosion (Figure 4). The harbor structures that define the northern boundary of the coastal sectors disrupt the natural sediment transport, leading to significant shoreline retreat downdrift of these structures. Based on the findings of Lira et al. [22], between 1958 and 2010, the shoreline in the stretch Barra-Vagueira retreated at a rate of 3.74 m/year while in Figueira da Foz-Leirosa, the retreat was 1.27 m/year. To mitigate effects of erosion and protect people and goods, several coastal interventions have been implemented, relying on the construction of costal structures (groins and longitudinal revetments). However, in recent years, the focus of coastal management is to mitigate the sediment deficit on the coastal system, performing artificial nourishments with sediments from dredging activities in the harbors [23].

### 4.1. Costs and Benefits

A cost–benefit analysis of sand bypassing systems is presented for the coastal sectors Barra-Vagueira and Figueira da Foz-Leirosa, where sand bypassing systems are commonly referred as a possible solution to mitigate the erosion effects that occur downdrift of the harbor structures. Firstly, the projections of costs and benefits for both study areas are presented in a 30-year life cycle. Subsequently, the costs and benefits are weighted, enabling an assessment and discussion of the physical and economic feasibility of the interventions to mitigate the erosion effects.

In order to determine the economic feasibility of the sand bypassing systems, the evolution of costs and benefits was obtained for the reference scenario. This reference scenario corresponds to the maintenance of the current policy of coastal erosion mitigation, which involves implementing nourishment interventions downdrift of the harbor breakwaters using sediments obtained from dredging activities in the harbors. In the presented study, all the economic analysis was developed considering a discount rate of 2%.

#### 4.1.1. Costs

The costs estimation of the sand bypassing systems was derived from the studies presented by Coelho et al. [8]. The costs of the sand bypassing systems were based on a review analysis of the costs of the systems implemented in other coastal areas around the world and in the experience of the authors in the development of coastal works in Portugal and Australia. Australian experts were also contacted, which brought for the

study the knowledge about the sand bypassing system implemented in the Gold coast, Australia, helping in identifying and quantifying all the system costs [24–27]. Thus, in this study, the total 30-year lifecycle of the bypass solution was considered, which includes the construction phase, year zero (preparatory work, implementation of pile-supported platforms, construction of technical buildings, equipment and piping); exploration phase, from years 1–30 (dredging operations, electricity and operational team); maintenance works, from years 1–30 (assumed to be 2% of the construction cost) and the dismantling phase, year 30 (assumed to be 5% of the construction costs) [28,29]. Additionally, the economic burden of maintaining the current policy of coastal erosion mitigation measures was identified (reference scenario) by analyzing the historical data of harbor dredging and deposition operations carried out in the study areas. Table 4 summarizes the costs estimation for both study areas, considering the reference scenario and the implementation of a fixed sand bypassing system.

**Table 4.** Summary of costs estimations (M€) for both study areas (based Coelho et al. [8]).

| | | Life-Cycle Costs | | | | Total Costs | |
|---|---|---|---|---|---|---|---|
| | | Construction | Exploration | Maintenance | Dismantling | Constant Prices | Updated Prices (2%) |
| Aveiro | Reference Scenario | 0.0 | 24.8 | 0.0 | 0.0 | 24.8 | 18.5 |
| | Bypassing | 22.2 | 54.6 | 12.7 | 1.1 | 90.6 | 73.9 |
| Figueira da Foz | Reference Scenario | 0.0 | 30.0 | 0.0 | 0.0 | 30.0 | 22.4 |
| | Bypassing | 18.1 | 43.0 | 10.3 | 0.9 | 72.2 | 58.9 |

The costs estimations suggest that maintaining the current practice of performing nourishments with sediments from maritime activities over the next 30 years will result in an economic burden of 18.5 M€ in Aveiro and 22.4 M€ in Figueira da Foz. The implementation of a sand bypassing system results in a total cost of 73.9 M€ in Aveiro and 58.9 M€ in Figueira da Foz at the end of the project's lifecycle. According to Coelho et al. [8], the factor that contributes to the higher costs of the bypassing system in Aveiro is the longer length of the sediment discharge circuit compared to the system in Figueira da Foz.

4.1.2. Benefits

Following the methodologies presented in Coelho et al. [8,15], the benefits of the sand bypassing systems were determined by combining the monetary values of the coastal territories with the systems' capacity to mitigate territory loss. The monetary value of the coastal stretches was determined considering land uses and benefit transfer, which involved utilizing land use cartography [30] and ecosystems values provided by Constanza et al. [18]. Both study areas were divided into 1 km stretches, and for each stretch, the monetary value and the impact of the bypassing system were quantified (Table 5).

**Table 5.** Monetary values of the coastal territories based on the territory division into stretches presented in Figure 4 (€/m$^2$/year), considered in the benefits estimates, function of accretion and erosion areas.

| | 1 | 2 | 3 | 4 | 5 | 6 | 7 | 8 | 9 | 10 | 11 | 12 |
|---|---|---|---|---|---|---|---|---|---|---|---|---|
| Aveiro | 38.0 | 21.2 | 18.8 | 29.2 | 6.0 | 4.6 | 2.4 | 0.4 | 3.9 | 29.7 | 8.0 | 0.5 |
| Figueira da Foz | 29.9 | 33.4 | 12.2 | 11.4 | 0.4 | 7.1 | 0.3 | 2.5 | 1.5 | 18.9 | 5.3 | 0.3 |

Avoided territory losses were estimated by comparing the annual shoreline positions between the reference scenario (without the bypassing system) and the scenario considering the implementation of a bypassing system. The areas of territory were determined

by utilizing projections of the shoreline positions obtained from the one-line numerical model LTC [31–33]. The initial step involves defining the model setup for each study area, which was accomplished by establishing a numerical domain using the bathymetry and topography of each study area. The numerical domain of each study area was defined with an extension of 12 km alongshore. In the calculation domain of each study area were included the existing coastal structures. In Aveiro, there were 19 coastal structures (8 groins and 11 longitudinal revetments), and in Figueira da Foz, there were 14 (8 groins and 6 longitudinal revetments). The wave parameters used as input for the model were obtained from the wave series generated as part of the MarRisk research project [34]. The longshore sediment transport in the model was defined to be calculated according to the CERC formula [19].

The model setups were calibrated to ensure accurate reproduction of the shoreline evolution in the study areas. This calibration process involved comparing the numerically obtained shoreline change rates with values suggested in the bibliography. Once the calibration was completed, the evolution of the shoreline position for the reference scenario was determined. Subsequently, the evolution of the shoreline position was evaluated by considering the sand bypassing system scenarios. To simulate sand bypassing systems in the LTC model, the users specified the hourly transposed volume and the locations of the outlets.

### 4.2. Intervention Scenarios

Based on the results of the Feasibility Study of Sand Bypassing Systems for Aveiro and Figueira da Foz Inlets [8], the bypassing systems were pre-designed to transpose a volume of $1 \times 10^6$ m$^3$/year through two outlets. The outlet locations follow the recommendations provided by Coelho et al. [8], and they are shown in Figure 4 through arrows. To evaluate the system's performance in relation to the volume of sediments transposed for each outlet, the benefits of the sand bypassing systems were obtained for two operational scenarios:

- AlternativeA (50/50): the shoreline position over time was obtained by considering that each outlet transposes half of the total volume of sediments;
- Alternative B (20/80): the shoreline position over time was obtained by considering that the northern outlet transposes 20% of the total volume of sediments while the other transposes 80% of the total volume.

### 4.3. Sand Bypassing Systems Feasibility

Table 6 provides a summary of the physical and economic balance of the reference scenario at different years. Based on the obtained values, it is expected that the shoreline position will continue to retreat over a 30-year time horizon, resulting in a loss of approximately 40 ha in Aveiro and 43 ha in Figueira da Foz. The total economic loss, which includes the value of territory loss, in Aveiro is estimated to be approximately 83 M€ while in Figueira da Foz, it is estimated to be around 65 M€.

**Table 6.** Physical and economic balance of the reference scenario at different years.

| | Aveiro | | | Figueira da Foz | | |
|---|---|---|---|---|---|---|
| **Year** | **10** | **20** | **30** | **10** | **20** | **30** |
| Area lost (ha) | 27.05 | 34.71 | 40.17 | 14.52 | 30.57 | 42.67 |
| Value of the area lost (M€) | 17.77 | 41.41 | 64.36 | 14.30 | 30.09 | 42.39 |
| Investment (M€) | 7.41 | 13.49 | 18.48 | 8.98 | 16.35 | 22.40 |
| Economic balance * (M€) | −25.18 | −54.90 | −82.83 | −23.29 | −46.44 | −64.79 |

* Values updated to the year 0, considering a discount rate of 2%. M€ means million euros.

Sand bypassing systems allow to mitigate the loss of territory over time in both study areas. As shown in Tables 7 and 8, the physical impact indicates that in Aveiro the system reduces the area lost by approximately 33 ha in Alternativen A and 45 ha in Alternative B.

In Figueira da Foz, the bypassing system enables the reversal of the erosive trend because the physical impact of the bypassing (109 ha) is higher than the area lost in the reference scenario (42.67 ha).

**Table 7.** Evolution of accumulated values of physical impact (non-lost area), costs and estimated benefits for Barra-Vagueira coastal sector.

| | Year | 0 | 10 | 20 | 30 |
|---|---|---|---|---|---|
| Alternative A | Physical impact (ha) | 0 | 13.48 | 22.80 | 32.95 |
| | Total costs * (M€) | 22.24 | 44.89 | 60.57 | 73.87 |
| | Total benefits * (M€) | 0 | 14.56 | 34.31 | 47.74 |
| Alternative B | Physical impact (m$^2$) | 0 | 15.73 | 30.68 | 44.94 |
| | Total costs * (M€) | 22.24 | 44.89 | 60.57 | 73.87 |
| | Total benefits * (M€) | 0 | 12.40 | 33.30 | 51.39 |

* Values updated to the year 0, considering a discount rate of 2%. M€ means million euros.

**Table 8.** Evolution of accumulated values of physical impact (non-lost area), costs and estimated benefits for Figueira da Foz-Leirosa coastal sector.

| | Year | 0 | 10 | 20 | 30 |
|---|---|---|---|---|---|
| Alternative A | Physical impact (ha) | 0 | 46.86 | 77.18 | 108.35 |
| | Total costs * (M€) | 18.08 | 35.65 | 48.19 | 58.86 |
| | Total benefits * (M€) | 0 | 61.25 | 184.92 | 333.00 |
| Alternative B | Physical impact (ha) | 0 | 47.83 | 82.73 | 109.79 |
| | Total costs * (M€) | 18.08 | 35.65 | 48.19 | 58.86 |
| | Total benefits * (M€) | 0 | 48.57 | 138.38 | 229.24 |

* Values updated to the year 0, considering a discount rate of 2%. M€ means million euros.

However, despite the positive impact of the systems in mitigating erosion in both study areas, the economic performance results indicate that the systems are only economically viable in Figueira da Foz. Based on the values presented in Table 6 and the evolution of the economic parameters NPV and BCR shown in Figure 5, it can be observed that in Aveiro, the total costs of the system (74 M€) are not fully compensated by the total benefits (approximately 48 M€ for Alternative A and 51 M€ for Alternative B) over a 30-year period (NPV < 0 and BCR < 1).

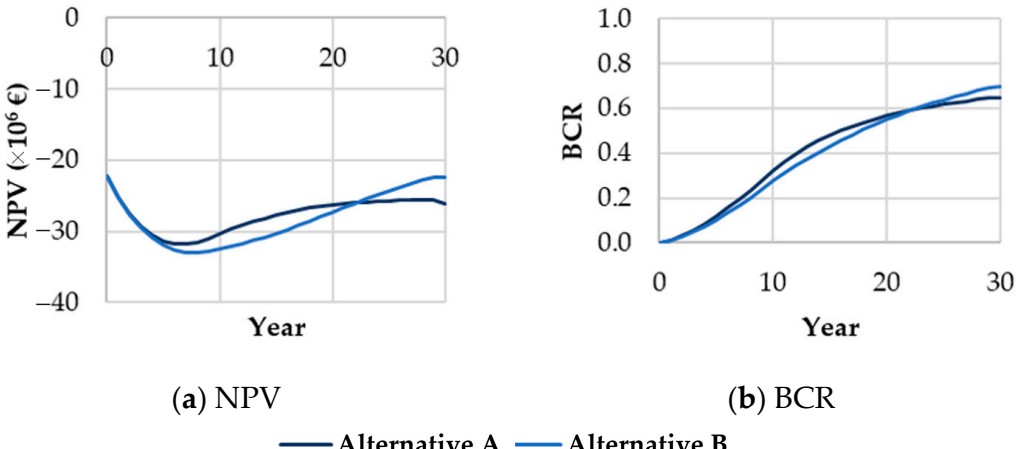

(**a**) NPV  (**b**) BCR

——— Alternative A  ——— Alternative B

**Figure 5.** Evolution of the economic parameters NPV and BCR for the bypassing system in the Barra-Vagueira coastal sector, considering the two alternative scenarios for benefit estimation.

In Figueira da Foz, the evolution of the economic parameters NPV and BCR demonstrates that the systems are economically viable (NPV > 0 and BCR > 1 after some time).

The break-even year is achieved in year 7 for Alternative A and in the year 8 for Alternative B (Figure 6). By the end of the defined project lifecycle, the total costs of the systems (58 M€) are fully covered by the benefits (333 M€ for Alternative A and 229 M€ for Alternative B).

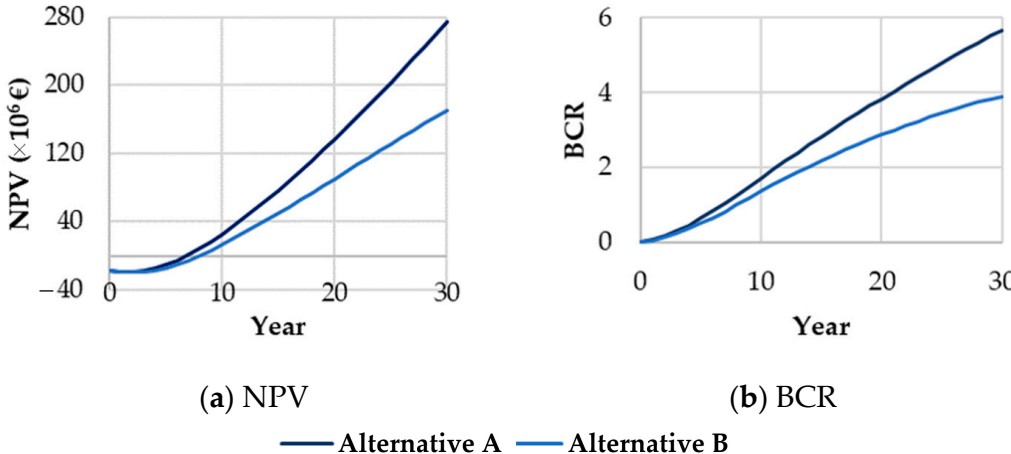

(**a**) NPV            (**b**) BCR

—— **Alternative A** —— **Alternative B**

**Figure 6.** Evolution of the economic parameters NPV and BCR for the bypassing system in the Figueira da Foz-Leirosa coastal sector, considering the two alternative scenarios for benefit estimation.

The differences in the results obtained for Barra-Vagueira and Figueira da Foz-Leirosa stretches can be justified by two main aspects. The Portuguese West coast exhibits some variability in longshore sediment transport. In fact, the longshore sediment transport was estimated higher in Aveiro compared to Figueira da Foz (996,000 m$^3$/year in Aveiro and approximately 746,000 m$^3$/year in Figueira da Foz), which will result in a better physical performance of the bypassing system at Figueira da Foz-Leirosa [8]. Furthermore, the greater gains in territorial area in the Figueira da Foz-Leirosa sector, induced by the transposition systems, are located in areas with higher monetary value, thus enhancing the benefits of coastal protection.

*4.4. Sensitive Analysis*

To test the uncertainty of the variables involved in the cost–benefit analysis, the economic feasibility of the sand bypassing systems was evaluated for a set of scenarios that assessed the impact of the following parameters: value of coastal protection, which includes the non-lost territory areas and territory value (the estimated value for benefits in the reference scenario was increase or decrease by 50%); the discount rate, testing the rates of 0% and 4% and the maintenance costs of the bypass system (the annual value considered in the reference scenario was increased by 5%). Table 9 provides a summary of the net present value obtained from the sensitivity analysis conducted.

**Table 9.** Net present value (M€) at the end of the life cycle of the project.

|  |  | Reference Scenario | Benefits (+50%) | Benefits (−50%) | Maintenance Costs (5%) | Discount Rate | |
|---|---|---|---|---|---|---|---|
|  |  |  |  |  |  | 0% | 4% |
| Aveiro | Alternative A | −26.13 | −2.26 | −50.00 | −40.32 | −25.03 | −26.90 |
|  | Alternative B | −22.48 | 3.22 | −48.17 | −36.67 | −18.03 | −25.19 |
| Figueira da Foz | Alternative A | 274.14 | 440.64 | 107.64 | 262.60 | 413.34 | 185.28 |
|  | Alternative B | 170.38 | 285.00 | 55.76 | 158.84 | 256.61 | 114.62 |

Generally, the sensitive analysis shows the same trend of results, giving confidence on the performed analysis. As expected, the results obtained show that increasing the benefits of coastal protection leads to better economic performance of transposition solutions. In Aveiro, when considering Alternative B, the increase in benefits makes the sand bypassing

system economically viable. In Figueira da Foz, the results suggest that even if the benefits are overestimated by 50%, the results still indicate that the benefits of the system outweigh the costs.

The increase in maintenance costs by 5% results in a worsened economic performance of the system for each study area. However, this increase does not render the solution economically unfeasible for Figueira da Foz. The scenarios assessing the impact of the discount rate reveal that an increase in the parameter leads to a decrease in the economic performance of the system for both study areas. Conversely, a decrease in the discount rate improves the economic performance of the system.

## 5. Discussion

The work performed allows for some discussion on bypassing system feasibility when applied to coastal erosion mitigation. The generic situation presented shows the importance on defining some characteristics of the system (location and sediment flux volumes), which highlight the relevance of the COAST tool to make the assessment of costs and benefits easier [14,15,35]. The two applications to real coastal stretches show that one adequate solution in one place is not necessarily good in another coastal stretch. In fact, the sand bypassing systems reduce the coastal erosion. However, its feasibility depends on the economic performance. The difference in economic viability between the transposition solutions in Aveiro and Figueira da Foz is partly attributed to the value of the benefits obtained, as the benefits in Figueira da Foz are significantly higher than those in Aveiro. This difference can be observed in the results related to the physical impact of the systems on shoreline evolution, where the transposition systems result in a considerably greater territorial gain in the Figueira da Foz sector compared to Aveiro. Additionally, the higher physical benefits at Figueira da Foz are located in most valuable zones, which increase the economic performance [8].

Wave climate, sediments characteristics, longshore sediment transport rates, land use and ecosystems services values, bypassing system construction, operation and maintenance costs, etc. are examples of important aspects that easily vary from one location to another. The site specificity is highlighted in this study due to the differences in the results of both real coastal stretches. However, future projections need to be interpreted carefully, and thus, the sensitive analysis performed helps on the confidence of the feasibility at the Figueira da Foz coastal stretch. In spite of that, same additional research should improve the knowledge on the benefits obtained with the mitigation of coastal erosion not only due to the territory maintained, but also due to flooding and damages related to overtopping events that are avoided due to the robustness of the downdrift beaches [36,37].

It is important to note that current decisions regarding coastal erosion management are significantly influenced by economic factors [38]. Robust CBAs that identify the relative costs and benefits of management options may assist coastal local councils, public authorities and their consultants, helping to make informed choices about which the management option (or options) will provide the greatest net benefits to their community. In fact, a well-constructed CBA can provide an important contribution to the information council in its decision-making processes [39]. However, it is always needed to be aware that the assumptions to evaluate the benefits encompass at the same time economic, social, cultural and environmental aspects, which should adequately characterize the provided services of the territory.

Other uncertainties in the performed analysis are related with the sediments management policies and longshore sediment transport estimates. Some scenarios were evaluated but other could be explored. Prices on dredging and deposition are changing quickly and may affect economic projections, increasing or reducing the competitiveness of one solution to the other [40]. The coastal management policies in Portugal shifted to a more natural based approach, considering sediments, against the hard-coastal structures [23,41]. The work performed shows that this may represent a good decision but depends on the sediment transport capacity in each coastal stretch and the consequent nourishment needs.

## 6. Conclusions

A cost–benefit analysis compares the costs and benefits of an intervention, allowing for considerations to be made about the economic viability of a project. COAST facilitates the development of cost–benefit analyses of coastal erosion mitigation interventions, following a sequential and well-defined methodology that allows for comparing all the costs and benefits of a given coastal erosion mitigation measure in updated values throughout the lifecycle set for the project.

The performed work shows that sedimentary transposition solutions are technically possible, allowing for the transposition of the potential longshore sediment transport. The outcomes of the generic case study illustrate that making slight modifications, such as adjusting the system's location or transposed sediment flow, can result in significant enhancements. Related to real cases, the Aveiro and Figueira da Foz systems differ slightly due to the characteristics of the locations themselves, forcing greater transposition distances (and higher costs) in the Aveiro system. Despite the uncertainty and subjectivity naturally inherent in a cost–benefit analysis, it can be concluded that any of the technically viable transposition solutions for Aveiro are, in economic terms, compensatory in a time horizon of 30 years. However, at Figueira da Foz, the shoreline evolution promotes the recovery of lost territory, so technically viable transposition solutions are economically viable in a time horizon of 30 years, assuming the defined assumptions.

Considering the uncertainty, volatility, complexity and ambiguity of future projections, whether from the perspective of the climate, the economy or socio-political (dis)balances, it becomes increasingly urgent and important to design and develop appropriate models for managing coastal territories. Cost–benefit assessment tools and transparent governance models will be a growing need and are already replacing traditional models, bringing new technologies, new knowledge and, above all, new processes that promote planning, implementing and evaluating the intervention strategies to mitigate coastal erosion and adapt coastal zones.

**Author Contributions:** Conceptualization, C.C., M.L. and A.M.F.; methodology, C.C. and M.L.; software, C.C. and M.L.; validation, M.L. and A.M.F.; formal analysis, M.L. and A.M.F.; investigation, M.L. and A.M.F.; resources, C.C.; data curation, C.C., M.L. and A.M.F.; writing—original draft preparation, C.C., M.L. and A.M.F.; writing—review and editing, C.C., M.L. and A.M.F.; visualization, M.L. and A.M.F.; supervision, C.C.; project administration, C.C.; funding acquisition, C.C. All authors have read and agreed to the published version of the manuscript.

**Funding:** This research received no external funding.

**Institutional Review Board Statement:** Not applicable.

**Informed Consent Statement:** Not applicable.

**Data Availability Statement:** Not applicable.

**Acknowledgments:** The authors express their thanks to APA (Portuguese Environment Agency), to COSMO programme from APA, co-financed by POSEUR, all members of the team that developed the study and all public entities that cooperate with the investigation. The research leading to these results has received funding from the EEA Grants 2014–2021 (Blue Growth Programme).

**Conflicts of Interest:** The authors declare no conflict of interest.

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
