# Peer review of "A Cost–Benefit Approach to Assess the Physical and Economic Feasibility of Sand Bypassing Systems"

_jmse, doi:10.3390/jmse11091829_

Round 1

Reviewer 1 Report

Table 1 is over two pages; move to one page for clarity

line 247, Figure number is missing

line 364, 375 table number is missing

The references to Situations A and B, are ideally 'Scenarios' or 'alternatives''. Might be a good idea to update the manuscript to call these scenarios instead. It's not terribly important though. 

Reviewer 2 Report

In this article the authors used the COAST model to perform a cost-benefit assessment to evaluate and discuss the effectiveness of bypassing systems during their life cycle. The results show that sedimentary transposition solutions are technically possible, allowing the transposition of the potential longshore sediment transport

 The idea of performing a cost-benefit analysis of the economic and physical feasibility of sand bypassing systems is novel and necessary but has seldom been addressed.

 The study is well done, but the text needs to be proof-read by a native speaker, since I found several grammar errors. I am not a native speaker, so I did not work on this. The methods and results are written in the same section, which I found difficult to follow. I would recommend writing these in two separate sections.

The discussion would benefit from comparisons with other similar studies, to highlight the value of this study. The concluding remarks could be shortended.

Based on the above, and the many details that I have outlined below, I think that the ms. needs a major revision before being accepted for publication. Because the study is interesting and the results are good, my recommendation is to accept after major revision.

Specific comments follow:

Abstract

The abstract is OK, but does not clearly mention the results of the study. An abstract should contain the results and conclusions of a study. Not in this case.

Introduction

The introduction is very interesting and well organized.

Although it is not within the goals of the study, I would add a few lines mentioning that there are also environmental impacts of sand bypassing, so that the reader dose does not forget this potential problem.

Methods

I am not an expert in numerical modelling, but it seems to me that the methods are well described and are adequate.

L 373-375 some editing here is necessary, because the table numbers are missing. Other figure and table numbers are also missing in the text. Please correct. This omission makes it difficult to read and understand the article.

The methods are confusing, because they show some results in them. This is the case of Tables 5, 6, 7 and 8, lines 367-371; 375-378; etc. I think the results should not be part of the methods section.

Results

The article lacks a results section. The results are shown in combination with the methods, and this makes the study difficult to follow. For clarity, I think it would be much better to have two separate sections: methods and results.

In Table 7 and elsewhere M, is it million euros?

Discussion

I think the discussion could be strengthened by comparing the results with other similar studies. It needs a major rewrite.

It would be interesting to compare if cost-benefit results are site dependent in other locations or countries.

I think the authors should mention, at least briefly, the environmental costs of bypassing systems. How are natural ecosystems, plants and animals possibly affected by sand bypassing?

Conclusions

I found the discussion too long. The conclusions do not contain the main results of the study. This is necessary. The take-home message (which includes the results) should be clearly stated in the concluding section. The first paragraph reads more like an introduction than a discussion. I would delete this paragraph.

The text needs to be proof read by a native speaker and by the authors too. The numbers of figures and tables are missing throughout the text. I found some unclear sentences. 

Reviewer 3 Report

This article adresses a relevant issue in coastal engineering, which is feasibility of systems of sand bypass from areas of accretion to areas where erosion is observed. Although the text is quite clear and well writen, a few issues were detected. I believe this article is not ready to be published at this moment. If the authors perform the below proposed corrections, I think it may be accepted ofr publication in the future:

1) Sand bypass systems are usually implemented next to jetties, or other similar coastal structures. Observing Figure 2, it seems intuitive to imagine that the northern boundary of the model would represent a jetty like structure, since no sediment transport rate was imposed on this boundary. Whit this in mind, it would be also intuitive to suppose that the sediment bypass outlet would be positioned near by the northern boundary. Instead, it is actually positioned 1,0 km far from the northern boundary. It is not clear why this choice was made. I suggest the authors to justify the alternative adopted. Conceptually, it is not a problem. However, the observed erosion seems not very acceptable in a real project.

2) Lines 88 - 93:  This paper uses economic parameters in order to discuss the feasibility of the analysed interventions. However, no introduction on these parameters is presented. Instead of simply citing other works, it would be interesting to introduce in more details the economic parameters apllied

3) It is not clear which reference values were apllied to obtain the costs presented in Tables 1, 3, 4 and 5. Instead of simply referencing other works, I suggest the authors to present these reference values.

4) Lines 299, 319 and 374 : Tables and Figures are not correctly cited. The table' and figure' numbers do not appear.

Round 2

Reviewer 3 Report

The authors performed most of the corrections requested in the first round of revisions. For this reason, the revised text is now clearer than the first version of this manuscript.  

However, the followingl topics still remain unclear:

1) The authors included the text between lines 106 and 114, which explains the economic parameters applied. It would be interesting and more easy to understand if these parameters were presented also as formulae.

2) The costs presented in Table 4 were obtained from the work of Carvalho et al (correct?). Besides citing this work, it would be interesting to provide a brief description of how this values were obtained in the cited work.

3) The source of the costs presented in Table 3 seems still not clear. I suppose it is the same source used in Table 4, correct?

4) It is not clear what the "Impact" in Table 3 means, or how it was calculated.

5) I suggest the authors to include in the text a Table, presenting the assumed values of cost per area of accretion and erosion.

If the authors perform these corrections, I believe this article may be accepted for publication in JMSE. 

Author Response

Please consider the attached file.
